# Global Cross-Border Malaria Control Collaborative Initiatives: A Scoping Review

**DOI:** 10.3390/ijerph191912216

**Published:** 2022-09-26

**Authors:** Tichaona Fambirai, Moses John Chimbari, Pisirai Ndarukwa

**Affiliations:** 1School of Nursing and Public Health, College of Health Sciences, Howard College Campus, University of KwaZulu-Natal, Durban 4001, South Africa; 2Pro Vice Chancellor’s Office, Main Campus, Great Zimbabwe University, Morning Side Drive, Masvingo P.O. Box 1235, Zimbabwe; 3Department of Health Sciences, Faculty of Sciences and Engineering, Main Campus, Bindura University of Science Education, Chimurenga Road off Trojan Road, Bindura P.O. Box 720, Zimbabwe

**Keywords:** cross-border malaria, border malaria control, malaria collaborative initiatives

## Abstract

Malaria remains a global disease of public health concern. Malaria control collaborative initiatives are widely being adopted to reduce malaria burden by various countries. This review sought to describe current and past cross-border malaria control initiatives focusing on key activities, outcomes and challenges. An exhaustive search was conducted in Web of Science, PubMed, Google Scholar and EBSCOhost using the following key words: cross-border malaria control, cross-border malaria elimination, bi-national malaria control and multinational malaria control, in combination with Boolean operators “AND” and “OR”. Eleven studies satisfied the inclusion criteria for this review. The majority of collaborative initiatives have been formed within regional developmental and continental bodies with support from political leadership. The studies revealed that joint vector control, cases management, epidemiological data sharing along border regions as well as resource sharing and capacity building are some of the key collaborative initiatives being implemented globally. Collaborative initiatives have led to significant reduction in malaria burden and mortality. The majority of collaborative initiatives are underfunded and rely on donor support. We concluded that cross-border malaria collaborative initiatives have the capacity to reduce malaria burden and mortality along border regions; however, inadequate internal funding and over-reliance on donor funding remain the biggest threats to the survival of collaborative initiatives.

## 1. Background

Malaria remains a major public health concern with at least 240 million cases and over 600,000 deaths recorded globally [1]. Global efforts in malaria control have led to a significant reduction in malaria burden and mortality in the last two decades [1,2]. Africa and Asia contribute the largest proportion to the global malaria burden [2]. Global efforts in vector control, malaria treatment and diagnosis dating back to the 1950s have contributed to the significant reduction in malaria burden [3]. The Global Malaria Eradication Program (GMEP) [4] of the 1950s and 1960s, Roll Back Malaria (RBM) program [5] and the Global Fund for Fight Against Malaria, HIV and TB (GF) [6,7] have also contributed to the significant decline witnessed in the last two decades. On the backdrop of the significant decline in malaria burden and mortality, many countries have embarked on malaria elimination programs [8]. The World Health Organisation (WHO) has led efforts to achieve elimination goals through a recently developed Global Technical Strategy (GTS) (2016–2030) with the overall aim of achieving malaria elimination in at least 35 countries by 2030 [9].

Key obstacles to the achievement of the malaria elimination agenda include, among others, vector resistance to insecticides, emergence of parasite resistance to current malaria treatment medicines and border malaria [10]. “Cross-border malaria” or “border malaria” is defined as malaria which occurs along or across international boundaries [11,12,13]. Cross-border malaria is complex to deal with and has the potential to reverse the gains achieved in malaria burden reduction over the past two decades [13]. Border regions possess a higher malaria receptivity and vulnerability index due to favourable climatic and human factors [14]. Malaria control efforts in border regions have also been impeded by poor access to health services and infrastructure by border communities [11]. Mobile populations sustain persistent malaria infections and movement of parasites across borders [15]. Failure to factor human population movement was identified as one of the main contributors to the failure of the GMEP [4]. Human population movement has been shown to increase the movement of parasites from endemic to non-endemic areas [16,17]. Evidence from Brazil [12], Swaziland [18] and South Africa [19] has shown the significant role of imported malaria and border malaria on sustaining local transmission.

One of the significant calls under the GTS is for member states to deepen country-to-country and regional collaboration [9]. In response to this call, bi-national and multinational efforts have been formulated to deal with border malaria. Varied and uncoordinated interventions across border regions in endemic countries have resulted in an incline in malaria control interventions and coverages across states [13]. The differences in policies, treatment regiments and control interventions also complicates malaria elimination efforts along border regions [12]. Through the GTS, there has been a wider call for deepening country collaborative activities in dealing with border malaria as well as increasing political commitment towards malaria control [9,20].

Malaria control programs exerts a huge financial cost on any economy particularly medium- to low-income countries [21]. The majority of endemic countries lack the internal funding capacity to fund malaria control and elimination activities [22]. Therefore, there is a growing need to mobilise financial resources for malaria elimination through regional and global alliances [20]. In addition to resource mobilisation, collaborative initiatives have to be formulated to facilitate health data information sharing between affected countries. Evidence shows that integration and collaborative activities are increasingly being adopted as proven tools to aid malaria elimination [23]. It is clear from a historical perspective that no single formula exists to attain total malaria elimination [4,12]. Since the mid-1990s, various countries have forged collaborative efforts to combat border malaria in regions of Africa, Asia and the Americas. Due to the successful gains realised in malaria control over the past two decades, many countries are orienting their malaria control program towards elimination [24]. Despite the growing evidence showing the gains of cross-border malaria control initiatives and elimination, the road to eradication is hindered by technical, operational and financial obstacles [25].

Threats of malaria re-establishment [26] and imported malaria in regions where malaria was previously eradicated has buttressed the need for interstate collaboration. Increased global travel [27,28] and the emergence of artermisinin resistance in South East Asia [29] have heightened the need for collaborative efforts in malaria control. Minimal bodies of knowledge exists on cross-border malaria collaboration activities, success, challenges and their impact on border malaria. A previous study reported only on regional initiatives excluding bi-national initiatives [20]. The study did not report on the epidemiological and programmatic impact of the various collaborative initiatives. Taken together, the aforementioned issues suggest that there is a need to improve knowledge on bilateral and multilateral initiatives as well as to document their epidemiological impact. In this review, we sought to describe the various current and past collaborative initiatives that have been implemented across the globe. 

## 2. Methods

We conducted a scoping review of literature in Web of Science, PubMed, Google Scholar and EBSCOhost (Medline). The search considered articles published in English journals. The search used Boolean operations “AND” and “OR” with a combination of the following key terms: cross-border malaria control, cross-border malaria elimination, bi-national malaria control, multinational malaria control. Additional literature was also obtained through a snowballing technique using bibliographies of previously published articles and reference lists. After the initial search, duplicates were removed. The remaining articles were screened by title and abstract. The process leading to papers selected for full review was informed by the Preferred Reporting Items for Systematic Reviews and Meta-Analyses (PRISMA) (Figure 1) [30]. A further search was also conducted for initiatives identified in selected articles to enrich this review on the world wide web (WWW) using the Google search engine. However, initiatives identified through the world wide web search were not included in the final analysis as they did not satisfy the inclusion criteria.

### 2.1. Inclusion and Exclusion Criteria 

Studies were included in the review if they were published in an English peer reviewed journal and reported on a past or existing bi-national, tri-national, regional or continental cross-border malaria control collaborative initiative. Any literature that did not satisfy the inclusion criteria was excluded from the review.

### 2.2. Data Extraction

A data capturing template shown in Table 1 was used to extract the following information: author, year of publication, region/country, type of collaboration, summary of key activities, outcomes and challenges.

## 3. Results

A total of eleven (11) articles were included in this study from the initial yield of 1405 papers searched from Web of Science, Google scholar, PubMed and EBSCOhost (Medline). A total of 44 duplicates were removed from those that were screened. The review further removed 1340 papers which were deemed not necessary for this review (Figure 1). Table 1 summarises the 11 studies included in the final review. Of the 11 articles, 10 focused on initiatives within the African, Asia Pacific and Americas regions. One paper reported a partnership between a country and a continent. The six additional collaborative initiatives identified through the world wide web search are outlined in Table 2.

The collaborative organisation and setup were classified into four main categories: (i) Bilateral Initiatives, (ii) Tri-national Initiatives, (iii) Regional Initiatives and (iv) Continental Initiatives.

### 3.1. Bi-National Initiatives

Three papers reported on country-to-country arrangements: the Trans-Kunene, French Guiana–Brazil and Peru–Ecuador [39] collaborative initiatives. The Trans-Kunene Malaria Initiative (TKMI) involves Angola and Namibia in the northern border regions of Cunene, Cuando Cubango [37] whilst the Peru–Ecuador initiative was functional in the El Oro and Tumbes border regions of the two countries. The TKMI and Peru–Ecuador initiatives were created to conduct joint malaria programs along borders whilst Brazil and French Guiana created a partnership in 1996 to facilitate information sharing between the two states.

### 3.2. Tri-National Initiatives

Two papers reported on the tri-national collaborative partnership of the Lubombo Spatial Development Initiative (LSDI). This partnership was formulated by Mozambique, South Africa and Eswatini (formerly Swaziland) [31,34]. The Lubombo Initiative was operational from 1999 and terminated in 2011 [34]. During its operation, the LSDI initiative delivered joint malaria control programs within the Lubombo regions. After the collapse of the LSDI, the three countries formulated another successor initiative termed MOSASWA in 2015 [34] with an overall goal to guide the three countries’ transition from malaria control to elimination through joint support and control programs.

### 3.3. Regional Block Initiatives

Cross-border malaria control’s success is highly dependent on inter-country collaboration as parasites and vectors transcend geographical borders. Asian Pacific regional political leaders formulated the Asia Pacific Malaria Elimination Network Alliance (APMEN) through the Asia Pacific Leaders Malaria Alliance in 2008 [35]. Similarly, Southern African regional leaders through the Southern Africa Development Community (SADC) formed the Elimination Eight (E8) [36] initiative composed of nine member states in 2009. The E8 seeks to coordinate, guide and harmonise malaria control and elimination programs within the Southern African region.

### 3.4. Continental Initiative

The African heads of states formed the African Leaders Malaria Alliance (ALMA) in 2009 [40] to lead malaria control and resource mobilisation efforts on the continent. The leaders sought to strengthen the commitment of African political leaders towards malaria control and elimination goals underpinned by cross-border malaria control among member states. China has forged partnerships with over 50 African countries supporting training, supply of medicines, equipment, research and medical personnel [33].

### 3.5. Initiatives without Peer Reviewed Publications

A search on the WWW in combination with bibliographical referencing revealed six initiatives outlined in Table 2. The Amazon Malaria Initiative (AMI) regional collaborative initiative operates in South America, comprising seven countries [42]. The initiative has been in existence since 2001 with the goal of standardising malaria treatment guidelines, surveillance and control and prevention. Six countries within the Greater Mekong Delta sub-region operate a regional collaborative partnership which guides border malaria control and malaria elimination activities within the region. Furthermore, the partnership offers technical guidance to member states, building capacity of member state staff as well as strengthening regional disease surveillance [43]. Two initiatives were identified in the Southern African region namely the Zambia–Zimbabwe and Mozambique–Zimbabwe–South Africa cross-border malaria partnerships [44]. The two partnerships were created in 2013 and 2011, respectively. The two partnerships were created to achieve universal coverage, coordinate cross-border malaria programs, reduce malaria transmission and push towards the attainment of elimination status. Eight countries within the Sahel sub-region created an initiative with the sole objective to accelerate efforts towards elimination by 2030 through ensuring universal coverage and mobilisation of resources [45]. The Great Lakes region countries within the East Africa Community (EAC) formulated the Great Lakes Malaria Elimination Initiative (GLME) in 2019 [46], with the overall aim to attain malaria elimination. 

## 4. Key Collaboration Initiatives Activities

### 4.1. Information Sharing

Sharing of health data is critical in assessing performance of interventions as well as assisting in the evaluation of malaria epidemiological trends, vector bionomics and treatment efficacy. The E8, APMEN, French Guiana–Brazil partnership, TKMI, MOSASWA and LSDI initiatives created platforms for the sharing of technical and epidemiological data across member states [32,34,35,36,41]. The APMEN [35] grouping has also initiated the creation and sharing of geographical information systems (GIS) among its various member states. The French Guiana–Brazil initiative has prioritised the creation of information sharing platforms to allow health workers to make timely and evidence-based decisions. The information is shared via a cross-border malaria information management system jointly developed and managed by the two countries.

### 4.2. Joint Malaria Prevention and Control Programs

Border regions between states often share common climatological, vectoral and community social characteristics. These conditions are critical determinants of an area’s vulnerability and receptivity to malaria transmission. To ensure universal malaria control coverage and standardisation of intervention along border regions the LSDI [31] and MOSASWA [34], Peru–Ecuador [39] and TKMI [32] initiatives implemented joint case management, vector control and mapping along the common borders. The LSDI [31] and MOSASWA [34] have operated a joint indoor residual spraying (IRS) program using DDT, Bendiocarb and pyrethroids since 1999. In addition to IRS, the distribution of insecticide treated nets (ITN)/LLINS and case management guidelines were implemented in the two initiatives. In addition to vector control intervention the LSDI also implemented a vector monitoring program within the Lubombo regions through routine mosquito catches. The TKMI initiative between Namibia and Angola implemented a quasi-experimental program within the border regions of Kunene to assesses the effectiveness of LLINs in the prevention of malaria among children.

### 4.3. Resource Sharing 

Resource inadequacies significantly affect program performance. In order to close resource gaps in a common border region, health officials in the Peru–Ecuador border region of El Oro and Tumbes shared medicines [39]. Similarly, in the E8, MOSASWA and APMEN [34,35,36] initiatives, countries have embarked on resource sharing and common resource mobilisation to close program gaps.

### 4.4. Funding Mobilisation

Malaria control interventions require huge monetary resources to finance medicines, equipment and human resources. Countries within the Asia Pacific and Southern African regions have successfully used collaborative initiatives to secure funding from multilateral funders and GF to sustain malaria control and elimination programs. The APMEN [35], MOSASWA [34], LSDI [31] and E8 [36] initiatives are current beneficiaries of GF grants for malaria control and elimination.

### 4.5. Political Leadership Lobbying

Political leadership ownership and support is critical for the sustainability of any health intervention. The French Guiana–Brazil partnership, TKMI, LSDI and MOSASWA were created by countries under the auspices of economic development partnership agreements between states. In addition, APMEN and E8 alliances were created under the auspices of regional development blocks, namely the Asia Pacific Leaders Malaria Alliance (APLMA) [48] and SADC [44]. 

### 4.6. Infrastructure Development

Border areas in low resource settings with high malaria burden are often characterised by poor access to health services and low intervention coverages. These regions tend to be highly receptive and vulnerable to malaria transmission. The E8 initiative has embarked on the building of “health posts” along member state border regions [36]. Health posts are health facilities located along border regions offering basic malaria diagnosis and care services to border communities and mobile populations. The E8 group has created malaria situation rooms to allow for real-time monitoring of malaria trends within the Southern African region [36].

### 4.7. Capacity Building 

It is critical that human resources responsible for the functioning of collaborative partnerships be endowed with the requisite skills to carry out critical tasks related to malaria programming. The APMEN initiative has invested in technical meetings and training programs whilst the China–Africa [33,35] initiative has seen African health personnel receive training under China funded programs. The LSDI [31] and MOSASWA initiatives have invested in the sharing of expertise and knowledge on malaria among the South Africa, Eswatini and Mozambique health authorities [34]. The French Guiana–Brazil partnership developed a cross-border malaria information system (CBMIS) and ensured adequate training for all concerned staff users [41]. This ensured a successful rollout of the CBMIS.

### 4.8. Outcomes and Challenges 

The implementation of cross-border collaborative activities has resulted in significant reduction of malaria burden within endemic regions of Africa, Americas and South East Asia. Sharp et al. [32] and Maharaj et al. [31] reported a successful reduction in malaria burden along the borders of Mozambique, Swaziland and South Africa. Based on a joint implementation model TKMI recorded a significant reduction of malaria cases along the Angola and Namibia border regions of Kunene among children [32]. Similarly, a reduction in malaria cases was also reported in the Peru–Ecuador border regions of El Oro and Tumbes. In addition to the reduction in human malaria cases, the joint IRS programs also resulted in declining vector densities [31]. Joint efforts in operation research resulted in Peru–Ecuador’s health officials detecting chloroquine resistance among malaria cases [39]. The APMEN initiative has managed to grow technical expertise on border malaria and elimination within member states through technical meetings and training. Improved access to malaria prevention and care through the construction of health posts within the E8 region has been attributed with a 30% reduction in malaria incidences and a 40% reduction in deaths. Whilst notable successes have been recorded in various initiatives in this review, it is evident that major obstacles have also been encountered. One of the key challenges threatening the survival of initiatives is limited country internal funding capacity. Malaria is highly endemic in low resourced countries of Africa, Asia and America whose national gross products are overburdened with other developmental priorities. Despite commitments by member states within the LSDI initiative, the three member states failed to honour their financial commitments resulting in the termination of the initiative [31,34]. Despite signing joint program protocols, some member states within the E8 were reportedly reluctant to share data on the E8 platform and were also reportedly hesitant to adopt novel methods [36].

## 5. Discussion

We reviewed eleven (11) papers which reported on 10 cross-border malaria control collaborative efforts in regions in Africa, the Americas and Asia. These initiatives were formulated based on geographical proximity, common ecological zones and regional integration. The initiatives succeeded in reducing malaria mortality and burden across common border regions. These collaborative initiatives have also facilitated malaria surveillance health information, the conducting of joint malaria control programs and knowledge and technical expertise transfer between member states. Commitment of political leadership to the goals of elimination of cross-border malaria across the globe is evidently growing. Inadequate internal funding capacity and total reliance on donor support are major threats to border malaria control collaborative initiatives.

The bi-national and tri-national program outcomes of LSDI, MOSASWA, TKMI and Peru–Ecuador have shown the potential of alliances to reduce malaria mortality and morbidity along border regions. The reduction in malaria burden could be attributed to joint vector control and treatment programs along the border regions. IRS, LLINS and use of anti-malaria medicines have been shown to be effective tools in controlling malaria [1,34]. Joint malaria control programs were also complemented by the sharing of epidemiological data. The sharing of health data on human cases and vector distribution allows for effective evidence-based targeted intervention. Joint malaria control and case management programs could likely have increased access to vector control interventions as well as standardised care, thereby reducing the intervention gradient. Based on this evidence, bi-national and tri-national border malaria alliances should continuously be adopted as they possess the capacity to reduce malaria burden. Similarly, regional initiatives have also shown the ability to supress malaria transmission where implemented. Similar regional initiatives in the Amazon basin [42] were able to reduce transmission through combined standardised malaria control programs, surveillance, technical assistance, resources and data sharing.

The nexus between malaria and poverty has been established before [49]. Coincidentally, the majority of low-income countries (LICs) have the highest malaria burden globally which places a huge financial burden on their economies [21]. The single biggest threat to cross-border malaria control collaborative initiatives is inadequate financing as evidenced by the demise of LSDI [34]. The end of LSDI saw a significant malaria resurgence in Mozambique, Eswatini and South Africa [34] which necessitated the formation of a new initiative comprising the same countries. The resurgence of malaria after cessation of intervention programs remain a key threat, as shown historically in various settings [4]. Over-reliance on external donor support has shown a catastrophic negative effect post withdrawal of external donor support [50]. This calls for countries embarking on border malaria control to commit more resources from their health budgets as well as to create innovative and novel methods of funding. However, the ability to increase internal funding for the majority of LICs countries is ever threatened by the likelihood of global financial turmoil and poor gross national product (GNP) performance [51]. Countries embarking on cross-border malaria control programs should be aware that cross-border malaria elimination programs are costly and require long-term stable financing [21,25].

Increased commitment by political leaders for integrating efforts to combat border malaria and non-border malaria has gained prominence since the mid-1990s. The involvement of regional bodies as well as political leaders ensures long-term sustainability of initiatives [52]. The creation of E8, Sahel Malaria Elimination (SaME), Mekong Delta sub-region Malaria Elimination hub (MME) and APMEN initiatives under regional development blocks may point to increased awareness and proof of an effective malaria advocacy strategy in the past two decades. It is vital to note that this political commitment has not fully translated into tangible results on abilities of LICs to grow internal domestic funding for cross-border malaria control programs. Despite political leaders establishing initiatives through cooperative agreements and protocols, teething problems still remain, i.e., data sharing among member states. This calls for more engagement among state players to ensure adherence to protocols. 

Data sharing challenges reported by Raman et al. in E8 are a cause of concern [36]. The challenges have also been reported in another regional disease surveillance collaborative body in Africa [53]. National health data is a sensitive area and its sharing may be subjected to strict scrutiny and multiple layers of approval. A study in Asia revealed that different standards, languages, national structures and rules have been cited as major impediments to effective health data sharing [54]. Data sharing among member states is also likely to be impeded by technical, political, legal and ethical barriers [55]. It is essential for collaborative initiative member states to have binding agreements and clearly laid out transparent health data sharing protocols and standard operating procedures (SOPs) agreeable to all parties. Stakeholder involvement at the development stages of a collaborative initiative health information system may improve participation. The development of a CBMIS in the French Guiana–Brazil partnership has shown that an all-stakeholder involvement approach from onset to completion is key to acceptance and adoption of the health information system [41]. Training and staff development in health information management systems is vital as some officers may lack the confidence to share data on regional platforms due to fear of criticism and ridicule if they share suboptimal data. Initiatives encountering data sharing challenges may need to adopt best practices from programs such multiple Indicator Cluster Surveys (MICS), Demographic Health Surveys (DHIS) and Mekong Basin Disease Surveillance Network (MBDS) initiatives which have built robust and efficient data sharing methods among member states [55,56]. Timely data is vital for assessing the impact of interventions.

Malaria control interventions such ITNs/LLINs, IRS and case management play an important role in malaria burden reduction. Evidence from TKMI has shown that joint border malaria control interventions could be more effective if they incorporate risk communication and behaviour change activities. Health education and correct risk communication messages improve community ownership, and uptake of community programs. It is critical that all collaborative initiatives incorporate and elevate community engagement initiatives within core interventions such IRS, LLINS distribution and case management. Risk communication and community engagement have been shown to be beneficial as they have the ability to increase case detection, uptake of interventions, access to health and positive behaviour change [57]. 

Despite the various challenges of implementing cross-border malaria control, it is critical to note that cross-border malaria control and malaria elimination are public goods with positive spill over epidemiological, social and financial benefits to neighbouring countries [23]. The return on investment on embarking on cross-border malaria initiatives has been shown to be beneficial [21].

## 6. Limitation

The aim of this review was to identify all existing cross-border malaria initiatives and document their key activities, aims and objectives, compositions, outcomes and challenges. However, despite a comprehensive systematic search, only eleven (11) articles were identified and not all the articles documented the required variables. Three reviewed articles did not report both outcomes and challenges [32,35,38] whilst six (6) articles did not report on challenges encountered during the implementation of the respective initiatives [32,33,34,37,39]. This limited the effective comparative analysis of initiative activities and challenges and their impact on outcomes. Further studies are needed to document various initiative program outcomes and their challenges to inform future interventions. We are also cognizant of the implications of excluding non-English articles from the review. This may have excluded a number of studies in French, Portuguese and Spanish that would have met our selection criterial. Whilst the review identified other cross-border malaria control initiatives from the world wide web with no published articles, these were however not included in the selected articles for analysis.

## 7. Conclusions

Our review revealed the value of collaborative initiatives by various countries and regional bodies in reducing malaria burden along border regions. Therefore, cross-border malaria control initiatives should continually be adopted across the globe as an effective tool to attain malaria elimination and suppress border malaria. Malaria epidemiological data sharing, joint vector control, surveillance, case management and resource sharing are some of the critical key activities being undertaken by cross-border malaria initiatives globally. Unwillingness by member states to share data and adopt novel methods are key challenges in some regional initiatives. Evidently, there is growing political commitment to a cross-border malaria elimination agenda as evidenced by political leader platforms giving birth to some of the initiatives. The majority of collaborative initiatives in low resource settings are heavily dependent on donor support; therefore, there is a greater need to create sustainable funding mechanisms as well as grow domestic funding of border malaria control programs. 

## Figures and Tables

**Figure 1 ijerph-19-12216-f001:**
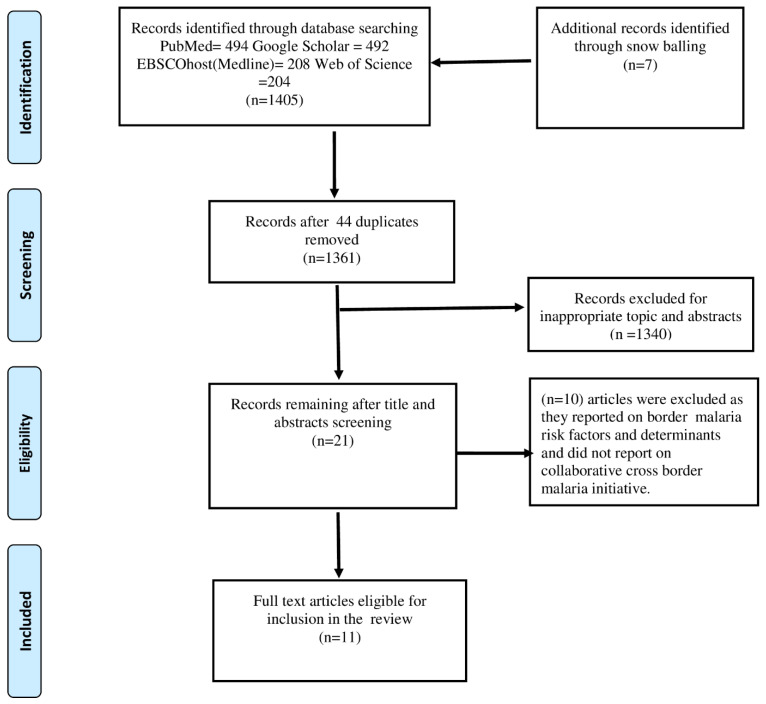
Preferred Reporting Items for Systematic Reviews and Meta-Analyses (PRISMA) flow diagram showing process of article selection.

**Table 1 ijerph-19-12216-t001:** Summary of articles and initiatives that were reviewed.

Author and Year	Name of Collaborative Initiative	Region/ Countries	Type of Collaboration	Operational Years	Aims/Objective(s)	Summary of Collaboration Key Activities	Outcomes	Challenges
Maharaj et al. (2016) [31]	Lubombo Spatial Development Initiative (LSDI)	Mozambique, South Africa, Swaziland	Tri-national	1999–2011	To accelerate socio-economic development in the region	The three countries shared technical expertise along the border regions of Kwa Zulu Natal, Eastern Swaziland and Southern Mozambique. A joint Indoor Residual Spraying (IRS) and case management program was implemented along the three nations common border region.	The initiative led to a significant overall reduction in malaria burden in the border regions. South Africa recorded a 99% decrease in malaria cases whilst Swaziland recorded a 98% reduction in malaria cases In Swaziland, malaria incidence declined from 28/100,000 pop in 2000 to <3/100,000 pop by 2001 In Mozambique, malaria prevalence declined from an average of 70% to less than 10% in (Zone 1–Zone 3). Manhica and Matola regions.	The financial burden for operating the initiative was huge. Financial commitment by member states did not materialise leading to closure of the initiative.
Sharp et al. (2007) [32]	Lubombo Spatial Development Initiative (LSDI)	Mozambique, Swaziland and South Africa	Tri-national	1999–2011	To accelerate socio-economic development within the three countries common border regions	The initiative instituted an Indoor Residual Spraying (IRS) program along the three country border regions. Entomological surveillance activities were conducted along the three countries’ common border regions. The initiative instituted routine monitoring of malaria cases along the border regions.	A significant reduction in *Plasmodium falciparum* prevalence was recorded from 60% in 1999 to 33% by 2005 within the Lubombo region i.e., in Mozambique project implementation zones (Zone 1–3), malaria prevalence declined in Zone 1 from 65% in 1999 to 4% by 2005. In Zone 3, prevalence declined from 70% in year 2000 to 33% in 2005. In Zone 2, prevalence declined from 69% in 2000 to 20% in 2005. Swaziland recorded a significant decline in malaria cases from 1395 in year 2000 to 200 cases by 2005. In South Africa, KwaZulu Natal district, malaria cases declined from 41077 in 1999 to 1771 by 2005, whilst in Mpumalanga province, malaria cases declined from 13656 in 1999 to 3099 by 2005 A significant reduction in vector abundancy and density was recorded in Mozambique attributable to the an effective IRS program i.e., *Anopheles. gambiae s.l.* catches declined from 5077 mosquitoes (pre-IRS) to 969 mosquitoes (post-IRS) *An. funestus s.l.* catches declined from 8830 (pre IRS) to 2107 (post IRS) Vector density declined from 14.9 (pre IRS to 0.2 (post IRS) for *An. arabiensis ss* Density for *An. funestus s.s.* declined from 26.3 (pre IRS) to 0.9 (post IRS) Sporozite index for *An. arabiensis s.s.* declined from 7.1 (pre-IRS) to 0.8 (post-IRS) *An. funestus s.s.* sporozite index declined from 1.2 (pre-IRS) to 0.02 (post-IRS)	Not indicated
Xia et al. (2014) [33]	China-Africa Initiative	China and Africa	Continental	1950s-present	To foster closer socio-economic China and Africa ties	China supports Africa medical services through training and supply of equipment. China provides academic exchanges and training programs for health workers and academics China deploys medical personnel to African countries China donates anti-malaria medicines to various African countries China has piloted malaria elimination programs in African countries. China supports the registration of pharmaceutical products.	China supports construction of thirty malaria research centres across thirty (30) across African countries China piloted malaria elimination programs in African countries	Not indicated
Moonasar et al. (2016) [34]	MOSASWA (Mozambique, South Africa and Swaziland)	Mozambique, South Africa and Swaziland	Tri-national	2015-Current	To accelerate transition from control to pre- elimination in Southern Mozambique and accelerate the transition from pre elimination to elimination in Eswatini and South Africa.	The initiative implements a joint support and expanded coverage IRS program. The partnership has scaled up drug based parasite clearance strategies and accelerated transition from pre-elimination to zero local transmission. The initiative mobilise resources and advocates for increased long term financing of malaria control and elimination programs The initiative has also created strategies targeted towards migrant and mobile populations	Not indicated	Not indicated
Gosling et al. (2012) [35]	Asia Pacific Malaria Elimination Network	Asia (Vanuatu, Solomon Islands, Bhutan, China, Democratic Republic of Korea, Indonesia, Malaysia, Philippines, Republic of Korea, Sri Lanka,	Regional	2009-current	Strengthen regional and multi sectoral collaboration around evidence based practises to reach malaria elimination goals.	APMEN designs and implement training programs and also promotes the use of Geographic Information Systems (GIS). The alliance has developed community engagement strategies for malaria elimination. The partnership provides funding for training program in member states The alliance holds annual technical meetings and produces a series of country cases studies	Not indicated	Not indicated
Raman et al. (2021) [36]	Elimination 8 (E8)	Mozambique, Zimbabwe, Malawi, Namibia, South Africa, Eswatini, Lesotho, Angola Botswana	Regional	2010-present	To strengthen regional collaboration in Southern Africa toward malaria elimination goals. To facilitate policy harmonisation, reduction of cross border transmission and mobilise additional resources	The initiative was created by Ministers’ of Health of SADC member states to enhance cross border malaria control initiatives and deployment of malaria health units at strategic points along the borders.	E8 funded the construction of thirty three (33) health posts in eight (8) SADC countries border regions. The health posts have been credited with a 30% reduction in malaria incidence and 40% reduction in malaria mortality. Through the efforts of the alliance malaria trends have been on a decline in certain Southern African countries. i.e. In South Africa, malaria cases declined from 30 000 in 2017 to 12 000 in 2019 In Namibia, malaria cases declined from 60 000 in 2017 to <3000 in 2019. The initiative has created malaria situation rooms The group keeps the malaria issue on the agenda at SADC summits. The initiative plays a pivotal role in strengthening political commitment for malaria E8 lobbied for increased domestic funding among member states	Member states lack internal domestic financing capacity to independently support malaria control programs. The initiative is totally funded by donors. There is ministerial and technical team policy misalignment Countries reportedly reluctant to share malaria epidemiological data on regional E8 platform despite signing protocols Countries within the alliance are reluctant to adopt new technologies and techniques brought through the initiative An existing overburdened health systems with endemic bottlenecks within the region
Khadka et al. (2018) [37]	Trans-Kunene Malaria Initiative	Angola and Namibia	Bi-national	2012-current	To enhance bilateral collaboration, joint malaria control between Angola and Namibia along the northern Namibia and Southern Angola border region	Is an arrangement between government of Namibia and Angola The initiative was formed to enhance cross border malaria control in the two countries border regions of Cunene-Cuando Cabango. The partnership facilitate sharing of technical scientific information between the two countries	A quasi-experimental intervention conducted by the partnership over two years utilising treated bed nets; led to a significant reduction of odds of malaria fever among children by 54% (aOR 0.46 95% CI: 0.29–0.73). Among children under two (2) years, the odds of fever were reduced by 71% (aOR 0.39 95% CI: 0.23–0.65). Among children over two (2) years, the odds of fever were reduced by 47% (aOR 0.53 95% CI: 0.30–0.65).	Not indicated
Kooma et al. (2017) [38]	Trans-Zambezi Malaria Initiative	Angola, Namibia, Botswana, Zambia, and Zimbabwe	Regional	2006-current	The collaboration aims to accelerate the reduction of malaria transmission among the border communities through implementation of coordinated cost effective malaria control activities	The initiative promotes cross border malaria collaboration as well as support SADC and E8 malaria elimination goals	Not indicated	Not indicated
Krisher et al. (2016) [39]	Ecuador–Peru Collaboration (Unofficial)	El Oro Region (Ecuador) and Tumbes Region (Peru)	Bi-national	1995-current	Collaborative was created unofficially by the two countries health officials to share epidemiological information	Local health officers created an unofficial cross border collaboration initiative for malaria control which resulted in sharing of epidemiological data, resources and conducting operational research	The initiative led to a significant reduction in malaria incidence in El Oro (Ecuador) region from 230 malaria cases per 10,000 pop. in 1999 to <10 cases per 10 000 pop. by year 2012 In Tumbes region (Peru) malaria cases declined from 1800 cases per 10,000 pop. in 1999 to <10 cases per 10,000 pop. by the year 2012 The collaboration between the two nation’s health officials led to the identification of Chloroquine resistance leading to its cessation as a drug of choice.	Not stated
Sambo et al. (2009) [40]	Africa Leaders Malaria Alliance (ALMA)	African Region	Continental	2009-current	To enhance and sustain African leaders commitment towards malaria elimination	The alliance is composed of African heads of states. The alliance seek to strengthen African leaders commitment to malaria control and elimination. The alliance also aims to strengthen cross border malaria control programs across African states.	Not indicated	Majority of malaria endemic countries lack domestic funding capacity to independently support own malaria control programs
Saldanha et al. (2020) [41]	French Guiana-Brazil	French Guiana and Brazil	South America	1996-present	Cooperative agreement created to improve health status of the two country’s common border region population	Regular sharing of epidemic data	The alliance created a harmonised cross border malaria information system (CBMIS) which improved access to data for all health officials and stakeholder The CBMIS assisted the two countries’ health authorities to timeously assess malaria epidemiologic dynamics in both space and time.	Limited health data access for health official in the two countries and different tools and terminology before adoption of CBMIS

**Table 2 ijerph-19-12216-t002:** Cross-border malaria initiatives with no peer reviewed articles.

**Name of Initiative**	**Region**	**Initiative Type**	**Country Composition**	**Operational Years**	**Aims and Objective(s)**
Amazon Malaria Initiative (AMI) [42]	Amazon Region	Regional	Brazil, Colombia, Ecuador, Guyana, Guyana, Peru, Suriname	2001-present	To standardise malaria prevention and control interventions through development of standard treatment guidelines, sentinel surveillance, trials and research.
Mekong Malaria Elimination (MME) hub [43]	Greater Mekong Delta sub-region	Regional	Laos, Cambodia, Vietnam, China (Yunnan Province), Myanmar Thailand	2017-present	To develop and coordinate partnerships and strengthen advocacy communication To provide leading technical support on cross country malaria projects and regional surveillance
Zambia Zimbabwe (ZAM-ZIM) [44]	Southern Africa	Bi-national	Zimbabwe and Zambia	2013-present	To strengthen cross border collaboration and coordination of malaria elimination along Zimbabwe and Zambia border.
Sahel Malaria Elimination Initiative (SaME) [45]	Sahel sub-region	Regional	Burkina Faso, Cape Verde, Chad, Mali, Mauritania, Niger, Senegal, Gambia	2018-present	To accelerate towards the attainment of malaria elimination goals by 2030. To scale up and sustain universal coverage of anti-malarial medicines and mobilizing financing for elimination. To fast-track the introduction of innovative technologies to combat malaria and develop a sub-regional scorecard.
Great Lakes Malaria Elimination Initiative (GLMEI) [46]	Central and East African region	Regional	Burundi, Democratic Republic of Congo, Kenya, Rwanda, South Sudan, Uganda, Tanzania	2019-present	To strengthen control and elimination of malaria in the Africa Great lakes region with main focus on cross border areas.
Mozambique Zimbabwe South Africa (MOZIZA) [47]	Southern Africa	Tri-national	Mozambique, South Africa and Zimbabwe	2011-Unknown	To achieve universal coverage of key malaria interventions To reduce transmission and eliminating malaria in districts which share borders in the three countries

## Data Availability

The data presented in this review is available in the public domain: Google Scholar, Web of Science, Medline and PubMed.

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
