# Peer review of "Global Cross-Border Malaria Control Collaborative Initiatives: A Scoping Review"

_ijerph, 2022, doi:10.3390/ijerph191912216_

Round 1

Reviewer 1 Report

The authors presented a descriptive summary of the key activities, outcomes and challenges of cross border malaria control initiatives. A systematic search was conducted in PubMed, Google Scholar and Ebscohost and there were only 10 studies that satisfied their inclusion criteria for further review. Joint vector control, case management, epidemiological data sharing and resource sharing were identified as the key activities. A significant reduction of malaria mortality and burden was considered as the major outcome. And inadequate internal funding and over-reliance on donor support were concluded as the biggest challenges for cross-border malaria control.

Overall, I believe this review has provided useful information for program designers, policy makers and stakeholders and some implications for future malaria control initiatives. However, there are several issues that need to be addressed.

Major issues:

  1. The key words and databases used for their exhaustive search seem incomplete. Other synonyms of “cross border malaria control” or “border malaria control” can be used as the alternative key words. For example, I searched Google Scholar using “binational malaria control”or “multinational malaria control” and found a relevant studies (Saldanha et al., 2020 JMIR Public Health and Surveillance) that is excluded from this review. Why the exhaustive search was not conducted in the Web of Science?
  2. Major findings lack support from data, especially when speaking of a reduction of malaria mortality/burden and a lack of funding, where is the data?

It would be helpful to provide the number of reduced mortality/economic burden, total funding and deficit of each control initiative, perhaps as column charts in the study. Otherwise these statements seem subjective.

  1. The discussion section seems excessive long and lacks of focus. I would recommend shortening it a little bit by highlighting your major findings and removing some unnecessary/irrelevant information.

Minor points:

  1. A lack of line number in the main text makes it difficult to locate the minor points.
  2. The sixth line of the abstract: Boolean operators “AND” and “OR”.
  3. The second line of the Background: 600,000 deaths
  4. Excessive empty space between two words occur throughout the manuscript, for example, the space between “contribute” and “the largest” in the fourth line of the background.
  5. The tenth and thirteenth lines of the second paragraph of the background: parasites.
  6. The Sixth line of the third paragraph of the background: … and control intervention also complicate…
  7. The third line of the fifth paragraph of the background: …global travel and emergence of artermisin…
  8. The last line of the first paragraph of methods: why initiatives identified through the world wide web search were not included?
  9. Figure 1: inclusion/exclusion criteria should be further clarified.
  10. Table 1: the challenge for reference 31(Sharp et al., 2007) was left empty

Author Response

Response to Reviewer 1 Comments

Major Comments

Number

Reviewer Comment

Authors response

1

The key words and databases used for their exhaustive search seem incomplete. Other synonyms of “cross border malaria control” or “border malaria control” can be used as the alternative key words. For example, I searched Google Scholar using “binational malaria control” or “multinational malaria control” and found a relevant studies (Saldanha et al., 2020 JMIR Public Health and Surveillance).   that is excluded from this review. Why the exhaustive search was not conducted in the Web of Science?

We thank the reviewer for the elaborate search of google scholar and for further improving on the key words that were used for searching the data bases. We have now improved our search and included Web of Science in our search as shown in revised PRISMA diagram (Figure 1) and Methods Section of the revised manuscript.

A revised search was done using the following key words: “cross-border malaria control”, “cross-border malaria elimination”, “bi-national malaria control”, “multinational malaria control” . The revised search only yielded one  eligible additional paper (already identified by reviewer) Saldanha et al., 2020) The paper was included in a revised analysis.

Major findings lack support from data, especially when speaking of a reduction of malaria mortality/burden and a lack of funding, where is the data?

The discussion section seems excessive long and lacks of focus. I would recommend shortening it a little bit by highlighting your major findings and removing some unnecessary/irrelevant information.

On epidemiological data:

The  missing malaria epidemiological data from articles that reported such data is now incorporated in the revised  results table ( Table 1). Only studies by Sharp et al.,2007 ( Lubombo Spartial Development Initiative) , Maharaj et al.,(2016) on (LSDI) Raman et al., (2021) on E8, Krisher et al.(2016) (Peru-Ecuador) and Khadka et al,(2018) Trans Kunene Malaria Initiative provided empirical data on epidemiologic impact of  respective interventions

On funding data:

Maharaj et al.,2017, Raman et al.,2021 and Moonsar et al.,(2016) merely mention existence of  funding gaps within initiatives, however the authors did not state or provide the quantum of monetary gaps.

Quoted below are some of the statements made by various authors concerning funding:

Maharaj et al.,(2016) concerning funding:

the opportunity to sustain the success the initiative was compromised as the result of financial constrains at country level……..,the financial resources that had been committed to the program by the country partners did not materials….As a result of the insufficient funds and problem in resource mobilisation. The LSDI was terminated in 2011”

Raman et al.,(2021): “.. in all E8 states except South Africa, the implementation of malaria is highly depended on access to external donor funding……. Over reliance on donor funding…… is further exacerbated by the fact that E8 is totally donor funded…”

The discussion section has been revised and redundant information has been removed.

Focus is now on key results/finding findings of this review.

(i) proven  ability of all initiatives (bi-national to regional initiatives) to suppress malaria burden and likely drivers of the success (joint interventions and information sharing)

(ii) financial threats to survival of initiatives in view collapse of a tri-national initiative as well as total reliance on external donor support in developing countries initiatives  

(iii) emerging political commitment towards cross border malaria

(iv) data sharing  challenges  

Minor Points

1

A lack of line number in the main text makes it difficult to locate the minor points.

Line numbers have been incorporated in the revised manuscript

2

The sixth line of the abstract: Boolean operators “AND” and “OR”.

The terms AND or OR have been replaced with “AND” and “OR”

3

The second line of the Background: 600,000 deaths

600 000 has been edited to 600,000 as shown in Line 33 pg.2: Background section

4

Excessive empty space between two words occur throughout the manuscript, for example, the space between “contribute” and “the largest” in the fourth line of the background.

The empty  spaces have been  removed  throughout the whole document in the revised manuscript

5

The tenth and thirteenth lines of the second paragraph of the background: parasites.

Plural of parasite has been inserted as shown in  line 56 pg. 2

6

The Sixth line of the third paragraph of the background: … and control intervention also complicate…

“Control interventions” has been inserted where  “control intervention” was appearing  as shown line  61 pg. 2

7

Figure 1: inclusion/exclusion criteria should be further clarified.

The included and exclusion has been revised and appear as shown Line 102  to Line 111-114 pg 3

8

Table 1: the challenge for reference 31(Sharp et al., 2007) was left empty

The authors (Sharp et.,2007) did not report any challenges in their article connected with implementation of the LSDI initiative.

Therefore “not indicated” had been  placed in “challenges” cell (Table 1)

8

The last line of the first paragraph of methods: why initiatives identified through the world wide web search were not included?

The initiative identified through the World Wide Web search were excluded from analysis because they did not meet the inclusion criteria as they lacked published peer reviewed  articles connected with their existence.

Save for the Amazon Malaria Initiative (AMI) and Mekong delta Malaria Elimination Initiatives (MME), the other four initiatives have no recorded program reports detailing key  activities,  outcomes and challenges. This further exclude them from the review. These initiatives in Table  2 were only included to enrich the article and to alerting the reader of existence of other initiatives. Result Section: Pg. 10

9

The third line of the fifth paragraph of the background: …global travel and emergence of artermisin…

The sentence has been corrected to “…. “Global travel and emergence of artemisinin resistance in …:” Background Section: Line 85 pg. 2

Reviewer 2 Report

This article reviewed and assessed the literature available on cross border malaria control collaborative initiatives. The review aimed to describe current and past cross border initiatives with a focus on their key activities, outcomes and challenges. Methods for this work included exhaustive searchers conducted on PubMed, Google Scholar and EBSCOhost using relevant key words, allowing for identification of 10 studies that satisfied the inclusion criteria. Review of the 10 studies identified that benefits of joint vector control, case management, epidemiological data sharing, resource sharing and capacity building across border regions towards reducing malaria burden and mortality. A total of nine initiatives were addressed: LSDI, China-Africa Initiatives, Mosaswa, Asia Pacific Malaria Elimination Network, E8, Trans Kunene Initiative, Trans Zambezi Malaria Initiative, Ecuador-Peru Collaboration and ALMA, which varied based on country composition (Bilateral, Trinational, Regional and Continental Initiatives). Underfunding and reliance on donor support was found to be a constant and global threat to cross border malaria collaborative initiatives, however, it was concluded that such initiatives have been successful towards reducing malaria burden across countries involved, and is an integral tool to attain elimination and suppress border malaria.

This review is focused on a vital aspect of global malaria control and management, which is cross border malaria control initiatives. There is an evident and renewed push for nations to come together, streamline activities, and share data and capacity development activities to ensure success towards elimination. As is outlined in the Global Technical Strategy (2016-2030), and mentioned by the authors in this review, it is aimed to achieve malaria elimination in at least 35 countries by 2030, which is proving to be difficult in some nations such as South Africa due to border and imported malaria. Therefore, such a review is a crucial publication to communicate the successes, challenges and overall importance of such initiatives to encourage further such activities, in addition to lobbying for continued and sustainable funding.

While the author’s review topic is highly relevant to the field and the authors have covered most aspects well, there is editing required to ensure there is consistency, clarity and key points of the review outcomes are expressed.  Some general comments are below:

  1. It is suggested to provide more detail on the exact successes of the initiatives as mention is briefly made of a few of the initiatives throughout, but direct comparison of the key activities relative to the outcomes is not clear, and this is the vital message to communicate to the reader. Comparison of the activities, limitations and successes between recognised initiatives should be highlighted to allow for the successful design of future initiative activities and for rapid adaptation.
  2. The structure of the review is well set out but there are some sentences that are unclear and aspects that are mentioned under sub-headings which do not correlate.
  3. The format of the tables needs to be addressed to ensure the text is clear, not extended and that borders and spacing are correct and maintained throughout.
  4. Figure 1 should have arrows and text boxes aligned.
  5. Consistency must be maintained in the language: e.g. Trinational versus Tri-national.
  6. Correct grammar must be provided for search engines, initiatives etc. E.g. Ebscohost versus EBSCOhost
  7. Examples of grammar and sentences that are suggested to be addressed throughout the text are included below with edits underlined:

  • Abstract, Page 1: Ebscohost corrected to EBSCOhost here and throughout text.
  • Abstract, Page 1: “Majority” to “The majority...”
  • Background, Page 1: line 2 “death” to “deaths”
  • Background, Page 1: “Key obstacles to achievement of the malaria elimination agenda...”
  • Background, Page 1: among other things: “among others”.
  • Background, Page 2: “…factor human population movement was identified as one of the main contributors to the ….”
  • Background, Page 2: “…resulted in an incline in malaria control interventions and coverages across states…”
  • Background, Page 2: “…policies, treatment regiments and control interventions…”
  • Background, Page 2: “The majority of endemic countries lack internal funding capacity to fund malaria control and elimination activities.” - Proof of this statement?
  • Background, Page 2: “Increased global travel and emergence of artemisinin resistance…”
  • Background, Page 2: “Minimal bodies of knowledge exist on cross border…”
  • Background, Page 2: “…knowledge on bilateral and multilateral initiatives…” - Remove lateral duplication.
  • Methods, Page 2: Correct EBSCOhost throughout.
  • Methods, Page 3: world wide web – “World Wide Web (WWW)”
  • Methods, Page 3: “Inclusion and Exclusion Criteria”- Sentence 1 requires rewording for clarity.
  • Results, Page 4: “(i) Bilateral initiatives, (ii) Trinational initiatives, (iii) Regional initiatives, and Continental initiatives” - either all in capital or not for initiative classifications.
  • Results, Page 4 - 7: Table 1:
    • Format appears to be stretched
    • Bordering is incorrect throughout
    • Appear to be inconsistent text and line spacing
    • A number of grammar edits required.
  • Results, Page 8: Table 2:
    • Spacing inconsistent with Table 1
    • “Southern Africa” = southern Africa throughout or Southern Africa if referring to UN region
    • Some grammar edits required.
  • Results, Page 8: Suggest to include whether “binational, trinational, regional or continental” in data Table 1 and 2.
  • Results, Page 8: “…three nation partnerships including Lubombo Spatial…”
  • Results, Page 8: “After the collapse of the LSDI, the three countries…”
  • Results, Page 8: “The leaders sought to strengthen commitment of African political leaders towards malaria control and elimination goals through underpinned by …”- remove either “through” or “underpinned by”.
  • Results, Page 8: “A single collaborative initiative operates in South America ….”
  • Results, Page 9: “The partnership offers technical guidance to members and capacity building, regional surveillance …” – Sentence does not make sense.
  • Results, Page 9: It is uncertain why Southern African initiatives listed under “Continental Initiatives” were included as they are considered Binational and Trinational?

  • Results, Page 9: (“Joint Malaria Control Programs”): “Case management with anti-malarial medicines reduces the level of parasitaemia and breaks the chain of transmission” – Unclear if this is a statement or referring to activity details of the LSDI and MOSASWA initiatives?
  • Results, Page 9: (“Resource Sharing”): “…countries have embarked on resource sharing and common resource mobilisation to close”
  • Results, Page 9: (“Funding Mobilisation”): Paragraph not clear as to if funding mobilisation took place in any initiatives. Only mention made of resource mobilisation. Not made clear where funding for initiatives is derived from or where planned future funding is aimed to be obtained from (i.e. private, governmental etc).
  • Results, Page 10: (“Infrastructure Development”): “The E8 initiative has embarked on the building of health posts along the member states…”
  • Results, Page 10: (“Outcomes and Challenges”): “…through construction of health posts within the E8 nine (9) countries border region..” – remove “nine” or replace with eight.
  • Results, Page 10: (“Outcomes and Challenges”): “Overburdened and weak health service delivery systems have also been identified…”
  • Results, Page 10: (“Outcomes and Challenges”): “Partnerships and initiatives usually result in discovery of innovative and cost effective ways of delivering services. Adoption of these novel methods was reported within the E8 initiative.” – It is not clear the methods or the innovations that were discovered? This statement should fall before challenges if it is a benefit.
  • Discussion, Page 11: “We reviewed ten(10) papers that reported nine (9) on cross border malaria collaborative efforts in Africa, Americas and Asia regions. – This sentence is not clear. Remove “on” following “nine (9)” and replace before.
  • Discussion, Page 11: “Discontinuing of LSDI exposed the critical role played by joints control efforts along border regions malaria as a significant increase in imported cases and malaria burden was reported in Mozambique, Eswatini and South Africa” – Sentence does not read well.
  • Discussion, Page 11: “It is critical to note that having adequate financial with pre requisite technical expertise is counterproductive” – Sentence does not read well, point not clear.
  • Discussion, Page 12: “The creation of E8, Sahel Malaria Elimination Initiative…”
  • Discussion, Page 12: “…impeded by technical barriers, political, legal and ethical barriers” – remove “barriers” after “technical.
  • Discussion, Page 12: “Involvement of other high ranking key non-health key…” – remove “key” duplication.
  • Discussion, Page 13: “…other global malaria control supporting partners bi-national and tri nation collaborative initiatives…” – lack of consistency. E.g. Binational versus bi-national, bi-national versus tri nation.
  • Discussion, Page 13: “Health education and correct risk communication messages improve community ….” – Remove “messages as “communication” is sufficient.
  • Discussion, Page 13: “…community health workers to increase case detection, access to health care(?) and improvement in behaviour change.”
  • Discussion, Page 13: “Due to social ties and economic humans will always be moving in search of economic and social ties, thus it is critical that strategies be created to address this aspect.” – Repetition and sentence is not clear.

Author Response

Response to Reviewer 2 Comments

Major Comments

Number

Comment

Author response

1

It is suggested to provide more detail on the exact successes of the initiatives as mention is briefly made of a few of the initiatives throughout, but direct comparison of the key activities relative to the outcomes is not clear, and this is the vital message to communicate to the reader. Comparison of the activities, limitations and successes between recognised initiatives should be highlighted to allow for the successful design of future initiative activities and for rapid adaptation.

 An exhaustive search was done for all successes reported in all selected all articles and the identified outcomes/successes are displayed in the revised Table 1.

Some authors did not outline fully program successes, whilst others did not report any outcomes/successes

Only 6/11 articles reported successes connected with respective initiative.

We summarised all the program that could be identified within selected articles.

Comparison of the various initiatives activities, limitation and outcomes has been done in the revised discussion section with key emphasis on:

 i.e. (I) impact of joint vector control, case management and information sharing on reduction in malaria morbidity and mortality

(ii) positive impact of all initiatives (bi-national, tri-national and regional initiative to supress malaria transmission).

(iii) challenges of data sharing in regional collaborative compared to easiness of data sharing in bilateral and tri-national initiatives.

(vi) Financial vulnerabilities of initiatives in low resources setting, over exposure to external donor funding and implications and way forward.

(iv) renewed political commitment within continental and regional developmental block towards malaria elimination and implications

Discussion Section:

Line 266 to Line

 284 Pg.20

2

The structure of the review is well set out but there are some sentences that are unclear and aspects that are mentioned under sub-headings which do not correlate.

The section with sub-headings have been revised to align text and headings.

Results section: Line 141 to 255

3

The format of the tables needs to be addressed to ensure the text is clear, not extended and that borders and spacing are correct and maintained throughout.

The tables have been re-formatted and appear as shown in Table 1 and  Table 2

4

Figure 1 should have arrows and text boxes aligned.

Figure 1 arrows and text have been aligned as depicted in the newly inserted Figure 1.

5

Consistency must be maintained in the language: e.g. Trinational versus Tri-national.

This has been rectified throughout the document

Where the word “Trinational” was appearing it has been substituted by “ Tri-national”

6

Correct grammar must be provided for search engines, initiatives etc. E.g. Ebscohost versus EBSCOhost

“Ebscohost” has been replaced by “EBSCOhost” through the manuscript.

Minor Points

1

Methods, Page 2: Correct EBSCOhost throughout.

Corrected as reflected in revised manuscript

2

Abstract, Page 1: “Majority” to “The majority...”

Corrected as shown in  Line 19

3

Background, Page 1: line 2 “death” to “deaths”

Corrected as shown in line 32

4

Background, Page 1: “Key obstacles to achievement of the malaria elimination agenda...”

Corrected as shown in line 45 pg. 2

5

Background, Page 1: among other things: “among others”.

Corrected as shown in line 45 pg.  2

6

Background, Page 2: “…factor human population movement was identified as one of the main contributors to the ….”

This has been corrected as suggested in line 55 pg. 2

7

Background, Page 2: “…resulted in an incline in malaria control interventions and coverages across states…”

Corrected as suggested in line 61 and 64 pg. 2

8

Background, Page 2: “…policies, treatment regiments and control interventions…”

Corrected as suggested in line 64 pg. 2

8

Background, Page 2: “The majority of endemic countries lack internal funding capacity to fund malaria control and elimination activities.” - Proof of this statement?

 A reference has been provided to support the statement:

Ren et al.,(2019) in their article:  “Greater political commitment needed to eliminate malaria”  identified domestic funding gaps in low incomes  endemic countries as one of the key challenges.

9

Background, Page 2: “Increased global travel and emergence of artemisinin resistance…”

“Artemisin” has been replaced by “artemisinin” as depicted in line 85 pg. 3

Background, Page 2: “Minimal bodies of knowledge exist on cross border…”

The sentence has been corrected as shown in line 86 pg. 2.

Background, Page 2: “…knowledge on bilateral and multilateral initiatives…” - Remove lateral duplication.

The duplicated word has been deleted: Line 92 pg.2

Abstract, Page 1: Ebscohost corrected to EBSCOhost here and throughout text.

“Ebsochost” has been replaced by “EBSCOhost” throughout the document

Methods, Page 3: world wide web – “World Wide Web (WWW)”

Methods, Page 3: “Inclusion and Exclusion Criteria”- Sentence 1 requires rewording for clarity.

Results, Page 4: “(I) Bilateral initiatives, (ii) Trinational initiatives, (iii) Regional initiatives, and Continental initiatives” - either all in capital or not for initiative classifications.

This has been corrected as suggested in Line 106

The inclusion and exclusion criteria has been reworded as shown in Line 111 to 115

The initiatives have been capitalised as shown in the main text in line 134 to 135 pg.5

Results, Page 4 - 7: Table 1:

Format appears to be stretched

Bordering is incorrect throughout

Appear to be inconsistent text and line spacing

A number of grammar edits required.

Results, Page 8: Table 2:

Spacing inconsistent with Table 1

“Southern Africa” = southern Africa throughout or Southern Africa if referring to UN region

Some grammar edits required

The format of Table 1 and Table 2 has been redone as shown in new results Table 1 and Table 2

Bordering has been corrected in the two Tables (Table 1 and Table 2) as shown in the revised manuscript.

Grammar errors within table have been corrected

Table 1 and 2 spacing has been harmonised. Single spacing has been used throughout the two tables as shown in new Table 1 and Table 2

Southern Africa has been adopted as the appropriate term throughout the document

Results, Page 8: Suggest to include whether “binational, trinational, regional or continental” in data Table 1 and 2.

Results, Page 8: “…three nation partnerships including Lubombo Spatial…”

A new column classifying an initiative whether it is a bi-national, tri-national, regional or continental has been added in revised Table 1.

The statement has been revised as shown in Line  of the manuscript

Results, Page 8: “After the collapse of the LSDI, the three countries…”

Results, Page 8: “The leaders sought to strengthen commitment of African political leaders towards malaria control and elimination goals through underpinned by …”- remove either “through” or “underpinned by”.

This sentence has been revised and now appear as shown in Line 174-175 pg. 10

This statement has been amended by deleting through and retaining “underpinned by” line 190 pg.12

Results, Page 8: “A single collaborative initiative operates in South America ….”

Results, Page 9: “The partnership offers technical guidance to members and capacity building, regional surveillance …” – Sentence does not make sense.

This sentence has been revised to “The AMI regional collaborative initiative operate in South America and is composed of seven (7) countries “in Line 199 pg.12

This sentence has been revised to “. The partnership offers technical guidance to members, build capacity of members states staff as well strengthen regional disease surveillance.” Line 200-205 pg. 12

Round 2

Reviewer 1 Report

The authors have made the suggested changes and additional searching and I believe the manuscript has been improved. However, some details may need further inspection and confirmation. For example, there are two typos/errors in the information for Ramen et al. (2021) in Table 1. First in Author and year column: wrong punctuation for “Ramen et. al(2021)” and second in Outcomes column: numbers are increasing but authors stated “declined from 30000 in 2017 to 120000 in 2019”.  

Author Response

3 Tembwe Road

Zengeza 5

Chitungwiza

Zimbabwe

25 May 2022

The Editor

International Journal for Environmental Research and Public Health

RE: RESPONSES TO REVIEWERS’ COMMENTS: GLOBAL CROSS BORDER MALARIA CONTROL COLLABORATIVE INITIATIVES: A SCOPING REVIEW: IJERPH-1702464

Firstly we would to thank you for the immense job you are doing in facilitating the review of our submitted paper. We do hereby submit our responses to Reviewer No.1 round 2 comments

We have put our responses to Reviewer No.1 comments in tabular form below

Major Comments

Number

Reviewer Comment

Authors response

1

The authors have made the suggested changes and additional searching and I believe the manuscript has been improved. However, some details may need further inspection and confirmation. For example, there are two typos/errors in the information for Ramen et al. (2021) in Table 1. First in Author and year column: wrong punctuation for “Ramen et. al(2021)” and second in Outcomes column: numbers are increasing but authors stated “declined from 30000 in 2017 to 120000 in 2019”

The punctuation error  on Raman et al. (2021) has been corrected as shown in revised Table 1.

Other observed typos were corrected as shown in the revised table.

The error on epidemiological data from Raman et al. (2021) article was revised. The  statement now reads In South Africa, malaria cases declined from 30 000 in 2017 to 12 000 in 2019 “

 Yours Faithfully

Tichaona Fambirai(Correspondence Author)
